# Structural comparison of homologous protein-RNA interfaces reveals widespread overall conservation contrasted with versatility in polar contacts

**Ikram Mahmoudi, Chloé Quignot, Carla Martins¤, Jessica Andreani** [ORCID] *

Université Paris-Saclay, CEA, CNRS, Institute for Integrative Biology of the Cell (I2BC), 91198, Gif-sur-Yvette, France

¤ Current address: MI-GSO PCUBED, Bâtiment Turcat II, ZAC Saint Martin du Touche, Toulouse, France
* jessica.andreani@cea.fr

**Data Availability Statement:** All relevant data are within the manuscript, its Supporting Information files, and on Zenodo at https://zenodo.org/doi/10.5281/zenodo.11126925, including a Jupyter

## Abstract

Protein-RNA interactions play a critical role in many cellular processes and pathologies. However, experimental determination of protein-RNA structures is still challenging, therefore computational tools are needed for the prediction of protein-RNA interfaces. Although evolutionary pressures can be exploited for structural prediction of protein-protein interfaces, and recent deep learning methods using protein multiple sequence alignments have radically improved the performance of protein-protein interface structural prediction, protein-RNA structural prediction is lagging behind, due to the scarcity of structural data and the flexibility involved in these complexes. To study the evolution of protein-RNA interface structures, we first identified a large and diverse dataset of 2,022 pairs of structurally homologous interfaces (termed structural interologs). We leveraged this unique dataset to analyze the conservation of interface contacts among structural interologs based on the properties of involved amino acids and nucleotides. We uncovered that 73% of distance-based contacts and 68% of apolar contacts are conserved on average, and the strong conservation of these contacts occurs even in distant homologs with sequence identity below 20%. Distance-based contacts are also much more conserved compared to what we had found in a previous study of homologous protein-protein interfaces. In contrast, hydrogen bonds, salt bridges, and π-stacking interactions are very versatile in pairs of protein-RNA interologs, even for close homologs with high interface sequence identity. We found that almost half of the non-conserved distance-based contacts are linked to a small proportion of interface residues that no longer make interface contacts in the interolog, a phenomenon we term "interface switching out". We also examined possible recovery mechanisms for non-conserved hydrogen bonds and salt bridges, uncovering diverse scenarios of switching out, change in amino acid chemical nature, intermolecular and intramolecular compensations. Our findings provide insights for integrating evolutionary signals into predictive protein-RNA structural modeling methods.

notebook to reproduce all figures in the manuscript and Supporting Figures, and to explore the data. Interactive exploration of the data is available by visiting https://bioi2.i2bc.paris-saclay.fr/django/rnaprotdb/.

**Funding:** The work was supported by grant ANR-18-CE45-0005 ESPRINet from Agence Nationale de la Recherche (to JA): https://anr.fr/Project-ANR-18-CE45-0005 The funders had no role in study design, data collection and analysis, decision to publish, or preparation of the manuscript.

**Competing interests:** The authors have declared that no competing interests exist.

## Author summary

Protein-RNA interactions are crucial to many biological functions and can play a role in diseases. We adopted a computational strategy to analyze and compare experimental 3D structures of protein-RNA interfaces. We first built a diverse dataset of 2,022 pairs of structurally similar protein-RNA interfaces, called structural interologs, and confirmed the existence of an evolutionary relationship in most of these interface pairs. We analyzed spatially close amino acid-nucleotide pairs across the interface, revealing that they are most often similar between interologs, even when the interfaces have strongly diverged. However, polar contacts such as hydrogen bonds are most often differently distributed between interologs, even in closely related interfaces. This finding highlights that spatial arrangement is more conserved than sequence in protein-RNA interactions and suggests principles guiding the evolution of these molecular associations. Our study has important implications for predicting protein-RNA interactions, both by providing useful rules for transferring contacts from a template with known structure to an interface of interest, and by paving the way for applying machine-learning techniques to integrate these patterns of contact conservation. This holds the promise of accelerating the identification of potential therapeutic targets and improving our molecular understanding for disease mechanisms mediated by protein-RNA interactions.

## Introduction

Protein-RNA interactions are crucial in many cellular processes, such as RNA metabolism, translation, DNA damage repair, and gene regulation [1,2]. They have also been implicated in numerous pathologies, such as cancers and neurological disorders [3]. Several studies of protein-RNA structures gave insights into possible pathological molecular mechanisms [4,5]. However, for many pathologies, the underlying mechanisms remain unresolved, leading to limitations in the proposed treatments [3,6,7]. Therefore, understanding protein-RNA interactions presents a major challenge in molecular biology. Detailed comprehension of those interactions is a crucial goal for medical and pharmaceutical purposes like drug design [6,7], which requires knowledge of the corresponding 3D atomic structures [2,8].

Even if the number of available experimental structures for protein-RNA complexes has greatly increased in the last decades, experimentally solving protein-RNA structures is still very challenging [2]. Only approximately 5,300 structures of protein-RNA complexes were available in the Protein Data Bank (PDB) in October 2023, compared to more than 200,000 entries overall, mostly proteins and homomeric protein complexes [9]. Therefore, computational tools for protein-RNA structural prediction and interface characterization have been the subject of dedicated research for several decades [10].

One major strategy for computational prediction of macromolecular interfaces is template-based prediction, which provides high-quality predictions for protein-protein complexes [11]. For protein-RNA interfaces, one pioneering study searched for structural interface similarity within a dataset of 439 non-redundant protein-RNA binary interfaces and identified a threshold of 25% for the minimum sequence identity based on alignment of the two proteins and the two RNAs, to identify structurally similar pairs of binary interfaces [12]. This study showed that above a sequence identity threshold of 30–35%, structural binding modes are similar and that many structurally similar complexes display low sequence identity. However, the study

did not investigate further how this conservation is enabled at the atomic scale and rather focused on template-based interface modeling.

When templates are not available or not detectable, interface modeling traditionally needs to resort to template-free docking. Free docking approaches most often consist of a sampling step generating many possible interface conformations, followed by a scoring step where these conformations are ranked [10]. Some of the most common scoring approaches rely on statistical potentials, deriving pairwise residue-ribonucleotide propensities [13–15] or hydrogen-bonding geometries [16] from known structures of protein-RNA interfaces. Other strategies involve coarse-grained force fields [17] or scores optimized by machine learning [18]. Protein-RNA free docking is more challenging than protein-protein docking due to the conformational flexibility of both protein and RNA partners [19] and the scarcity of high-resolution protein-RNA structures on which scoring methods can be trained. A hybrid strategy using both template-based and free docking improved protein-RNA interface structural predictions, especially in cases with low-homology templates [20]. Alternative methods, such as in the HAD-DOCK web server [21], include experimental restraints to guide the docking.

Evolutionary pressures apply to protein-protein interfaces to maintain interactions between partners [22]. Interface conservation and co-evolution signals between interface positions can be exploited to improve the structural prediction of protein-protein interactions in traditional docking [23], but also in global statistical methods exploiting covariation in multiple sequence alignments (MSAs) to derive the most likely direct contacts [24]. Many predictors of RNA-binding protein residues rely on machine learning using evolutionary information from Position Specific Scoring Matrices (PSSM) derived from MSAs or homology transfer from structural templates [25,26]. Alongside the propensity of amino acids (in particular positively charged residues) for binding RNA, evolutionary conservation derived from protein MSAs is one of the major features enabling the prediction of RNA-binding residues [27]. Co-evolutionary analysis can also indicate conserved RNA structures [28] and protein-RNA interfaces, with the caveat that these methods require large coupled MSAs and, therefore, are only applicable to a few bacterial protein-RNA complex families [29].

The recent release of powerful methods using deep learning algorithms to leverage information from MSAs, such as AlphaFold [30], AlphaFold-Multimer [31] and RoseTTAFold [32], has been a revolution for the structural prediction of proteins and protein-protein interactions. These methods have increased applicability compared to previous covariation-based models, thanks to the ability to exploit relatively small MSAs, and the success rates and precision of models have greatly increased compared to traditional docking methods. Recently, deep learning methods have been extended to predict protein-nucleic acid interface structures, notably with RoseTTAFold2NA [33] and, most lately, RoseTTAFold All-Atom [34] and AlphaFold3 [35]. These new methods encouragingly demonstrate the ability to learn joint parameters for diverse macromolecular interactions despite scarce protein-nucleic acid structural data and report high prediction performance for protein-protein complexes. However, the reported performance for protein-RNA structure prediction is much lower; for instance, the reported success rate based on interface local distance difference test (iLDDT) for a small test set of 25 protein-RNA complexes with low homology to PDB structures is 19% for RoseTTAFold2NA and 38% for AlphaFold3 [35].

An earlier study of the evolution of protein-protein interface structures [36] enabled us to identify conserved determinants that were subsequently useful to develop dedicated scoring functions that improved our predictive capacity [37–39]. Even in the current context of deep learning developments, understanding and leveraging evolutionary information remains crucial, and better protein-RNA interface structure prediction and scoring methods are still needed. Therefore, the structural analysis of protein-RNA interface evolution deserves special attention.

In the present study, we focused on the analysis of protein-RNA interface evolution, aiming to unravel the sequence and structural properties connected with interface conservation and plasticity. We identified 2,022 pairs of homologous protein-RNA interfaces with structurally similar experimental 3D structures and we used this unique dataset to perform a detailed analysis of how interface contacts are conserved between interologs. Abundant literature over more than twenty years [40–42] has defined important contact types for the energetics and specificity of protein-RNA interfaces: beyond atomic proximity, we also considered hydrophobic interactions, hydrogen bonds (H-bonds), and salt bridges, as well as π-stacking contacts. In this study, we highlighted the diverse conservation of these different contact types and the role of sequence divergence as a major determinant of contact conservation. We also explored the role of structural properties such as secondary structure and solvent accessibility in contact conservation.

## Results

### Identification of structural interologs

**Initial dataset of representative protein-RNA interfaces.** We first built a dataset of representative, high-resolution protein-RNA interface structures. From all experimental structures in the PDB, we retrieved the subset of entries containing at least one protein-RNA contact, where we defined contacts as amino acid/nucleotide pairs with a minimum heavy-atom distance below 5Å. From these 4,173 PDB entries, we extracted binary interfaces containing one protein chain in contact with either one RNA chain, or two base-paired RNA chains that we merged into one double-stranded chain. We removed interfaces with coordinates for only some backbone atoms (only Cα atoms on protein and/or only P atoms on RNA). On average, each PDB entry contains around 28 protein-RNA interfaces, although this number reflects a diverse range of situations (see supplementary results in S1 Text and S1 Fig). Because we aim for good-quality structures with well-defined atomic contacts, we applied resolution criteria, leading to 6,369 remaining interfaces, then we applied size criteria and obtained 3,419 interfaces. Finally, we applied clustering to remove strictly identical protein-RNA interfaces to avoid bias in our dataset. This pipeline (Fig 1) resulted in 977 representative protein-RNA binary interfaces.

**Structural comparison of interfaces to identify interologs.** Among these 977 representative interfaces, our goal was then to identify subsets of interface structural homologs (called structural interologs or interologs for short). We performed all-vs-all structural alignment of these 977 protein-RNA interfaces using TM-align [43], RNA-align [44], and MM-align [45] respectively for protein alignment, RNA alignment, and protein-RNA interface alignment (see Methods and Fig 2A). Of note, this structural alignment step is computationally costly, further arguing in favor of the clustering performed to obtain the 977 representative interfaces, as we avoided unnecessary interface comparisons. In principle, we should perform 476,776 (= 977 x 976 / 2) comparisons; however, we refrained from comparing pairs of interfaces that belong to the same PDB entry and have one chain in common, as these cannot be structural interologs (see Methods). Out of over 444,000 remaining possible comparisons, MM-align succeeded in aligning around 207,000 pairs of protein-RNA interfaces; in all other comparisons, MM-align did not simultaneously align the protein and RNA molecules and did not return the structural correspondence between both protein chains and both RNA chains. For the 207,326 successfully aligned pairs of interfaces, Fig 2B (gray points) shows the relationship between the interface TM-score provided by MM-align (measuring the interface structural similarity) and the minimum sequence identity based on sequence alignments for the two proteins and the two RNAs (using a local alignment, sequence identity weighted by alignment coverage, see

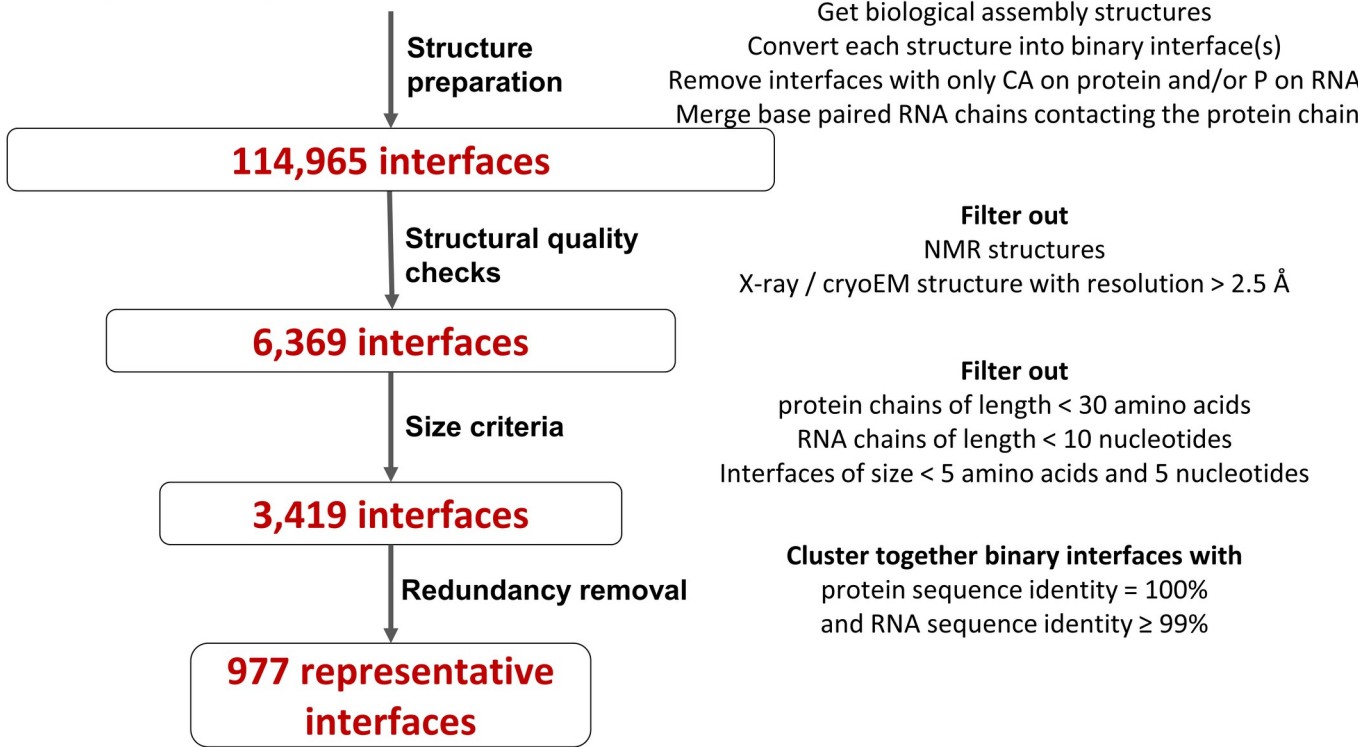

**Fig 1. Pipeline for the construction of the interface dataset.** The pipeline proceeds from all available protein-RNA complexes in the PDB (dated 21 February 2022) to the representative dataset of 977 interfaces used in this study.

Methods). This graph shows that above a sequence identity threshold of 25%, all pairs of interfaces display interface TM-scores above 0.5 (the standard threshold above which TM-align results are indicative of structurally similar folds), and a large fraction of those have interface TM-scores above 0.8. Conversely, the gray points in Fig 2B also display a densely populated region with interface TM-score above 0.5 but sequence identity below 25% (see also panel A in S2 Fig).

We next wanted to identify pairs of structural interologs from those pairs of interfaces with interface TM-score above 0.5. With structural visualization, we observed that interfaces may not be similar even when the interface TM-score provided by MM-align is high, e.g. because of low interface overlap (Fig 2C). On the other hand, when the RNA TM-score is low, the interfaces may be structurally similar, but the flexibility of RNA molecules might result in structures that are not overall superimposable (panel B in S2 Fig). Taking into account these observations, we did not use the RNA TM-score to define interologs; we chose cutoff values of 0.5 and 0.6 for interface and protein TM-scores, respectively, and we complemented the TM-score criteria with interface coverage criteria, whereby the interface overlap had to exceed 40% in both protein and RNA chains within the pair of interologs (Fig 2A). This resulted in a final set of 2,022 pairs of confidently assigned structural interologs, represented by the red points in Fig 2B, spanning a wide range of possible interface sequence identities (panel C in S2 Fig). Among these 2,022 pairs, 515 pairs have an RNA TM-score lower than 0.5 (panel D in S2 Fig). We computed interface root mean square deviation (RMSD) between interologs and verified that the vast majority of interologs have low RMSD values (99.9% below 6Å and 95.5% below 4Å,

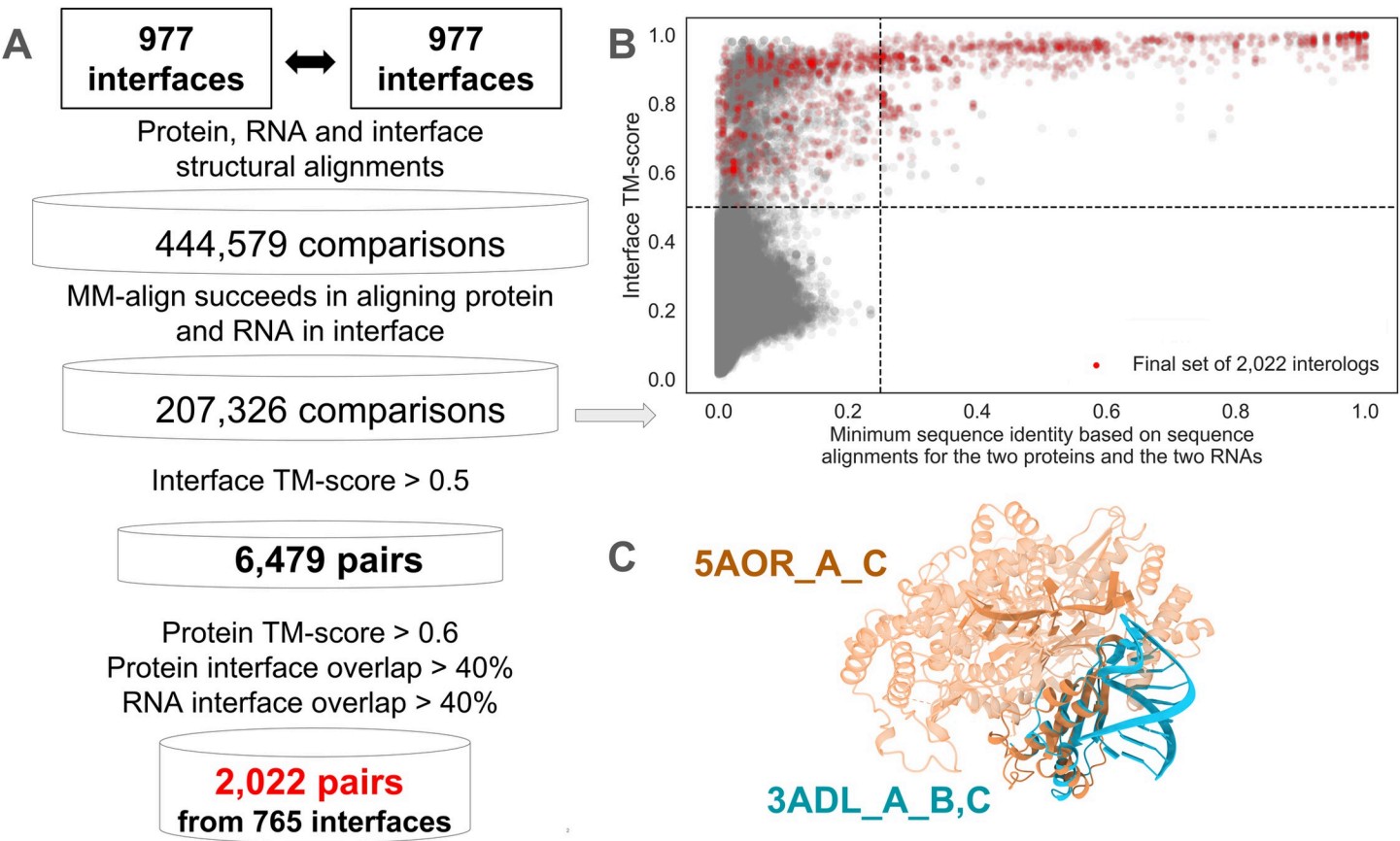

**Fig 2. Identification of pairs of structural interologs among the 977 interfaces in our dataset.** A: Pipeline for the identification of interologs from all-against-all structural alignment. B: Scatter plot of all-against-all interface TM-scores (y-axis) according to (x-axis) the minimum sequence identity within each pair of binary protein-RNA interfaces, weighted by alignment coverage. In the scatter plot, we excluded pairs of interfaces that MM-align failed to align (i.e. the scatter plot contains 207,326 gray points). Red points represent the final 2,022 pairs of structural interologs. Partial transparency was added to the scatterplot in order to reflect the density of points in different regions of the graph. C: An example pair of non-interolog interfaces aligned with MM-align where the interface TM-score is above 0.5 despite low interface coverage. 5AOR_A_C; 3ADL_A_B, C (associated protein TM-score 0.74, RNA TM-score 0.12, and interface TM-score 0.61) belongs to the intermediate set of 6,479 pairs. The two RNA chains do not bind in the same protein region so that the RNA interface overlap is 0% and this pair of interfaces is not retained in the final set of 2,022 structural interologs.

panel E in S2 Fig). Notably, 765 out of the 977 initial representative interfaces (78%) form the final 2,022 pairs and have at least one identified structural interolog in this process.

**Evolutionary relationship within pairs of structural interologs.** Our definition of structural interologs was so far purely based on structural similarity criteria. To assess whether the 2,022 pairs of structural interologs actually correspond to evolutionary homologs, we used evolutionary classifications to annotate our interfaces, namely ECOD [46] for protein chains and Rfam [47] for RNA chains (see Methods). We then assessed whether for each pair of interologs, the ECOD and Rfam classifications were consistent for the two structurally similar interfaces. For protein chains, 1,994 out of 2,022 pairs of structural interologs have ECOD annotations for both interfaces; out of these, 1,991 interolog pairs (99.8%) have at least one ECOD annotation in common at the T-group level (demonstrating homology and similar topological connections), and 1,803 interolog pairs (90%) have exactly the same ECOD T-group annotations. For RNA chains, 1,230 out of 2,022 pairs of interologs have Rfam clan annotation for both aligned interfaces and 100% of these have at least one clan in common (of which 99% have exactly the same clan annotation). 1,244 out of 2,022 pairs of interologs have

Rfam family annotation for both aligned interfaces, 57% of which (703 pairs) have at least one common Rfam family. The set of interologs with a common clan but no common family corresponds to more remote interfaces, e.g. archaeal vs. eukaryotic ribosomal RNA. This analysis shows that beyond structural similarity, our dataset of interologs also contains a vast majority of evolutionarily related interfaces, despite the homology relationship being very distant in a large fraction of the interologs, as evidenced by low sequence identities.

### Contact conservation analysis

**Definition of contact conservation and overall results.** For the 2,022 pairs of structural interologs, we compared the specific positions involved in each interface contact. Our strict definition of interface contacts, based on a minimum heavy-atom distance of 5 Å, is adapted to our need to precisely assess whether the close neighbors of an interface position are conserved in the interolog interface. Corresponding residues between interologs were defined from the interface structural alignment, as illustrated in Fig 3A. We assessed whether each interface contact occurring in our dataset was conserved in the structural interolog, irrespective of whether the nature of the amino acid/nucleotide varied (Fig 3B). To better represent the conservation of atomic contacts in a situation where the residue nature can vary, we weighed the conservation of each amino acid/nucleotide contact by the number of atomic contacts it contains (see Methods).

On average, 73% of these distance-based, atomic-weighted contacts were conserved among interologs. We also assessed weighted conservation for distance-based atomic contacts restricted to pairs of C atoms (a subtype we called apolar contacts). Contact conservation was 68% on average for these apolar contacts, overall quite high, albeit significantly lower than for all atomic contacts (panel A in S3 Fig). However, only 39% of the H-bonds, 31% of the salt bridges, and 36% of the π-stacking contacts were conserved on average, with all distributions significantly lower than for atomic contacts (panel A in S3 Fig). These differences in conservation levels might reflect that the interfaces are maintained in evolution through atomic and apolar contacts, while the specificity-driving H-bonds, salt bridges, and π-stacking interactions are more versatile.

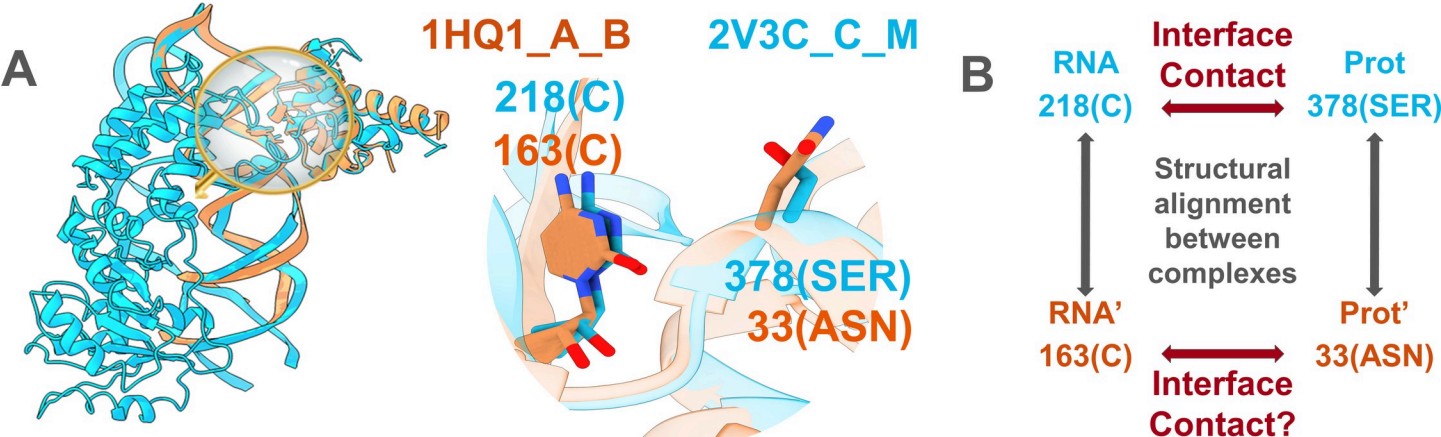

**Fig 3. Structural alignment of interologs and definition of contact conservation.** A: Structural alignment illustration for two structural interologs (1HQ1_A_B *Escherichia coli* interface in orange, aligned with 2V3C_C_M *Methanocaldococcus jannaschii* interface in cyan) sharing 62% minimum interface sequence identity. The complexes were aligned with MM-align and the zoomed region shows alignment details for two structurally aligned amino acid/nucleotide pairs. B: Illustration of the way contact conservation between interologs was assessed, based on structurally aligned positions.

Of note, H-bonds, salt bridges, and π-stacking interactions are much less abundant compared to atomic and apolar contacts (see supplementary results in S1 Text), and they require specific properties of amino acid side chains. These differences raise the question of how much the higher conservation levels for the more frequent atomic/apolar contacts can be driven by chance. Therefore, we introduced two versions of a random baseline, which involve either shuffling the interface residues or "resampling" i.e. drawing them from a distribution allowing us to maintain interface sequence identity with the considered interolog (see S1 Text for details of these baselines). In both baselines, we maintain the "scaffold" of the interface, i.e. which protein and RNA positions are involved and the distances between protein and RNA backbones, and we re-assign the contacts of all natures (atomic, apolar, H-bonds, salt bridges and π-stacking) formed in the shuffled/resampled interface. The contact conservation levels are similar in both baselines, and largely smaller than average conservation levels in the original interolog dataset (panels A and B in S4 Fig), underscoring the specificity of contact conservation between interologs. The smallest relative difference occurs for atomic contacts, highlighting that despite their overall higher conservation compared to other contact types, they are probably the least specifically conserved in interologs.

**Contact conservation depending on sequence identity and ribosomal/non-ribosomal character.** The 2,022 pairs of interologs display a wide range of sequence identities. We created four equally populated groups of interologs according to the minimum interface sequence identity based on the structural alignment between the protein-RNA interfaces: 0–19%, 19–34%, 34–60% and 60–100%. Interfaces involving a ribosomal protein form the majority of our dataset of interologs: we assigned 615 out of 977 interfaces as ribosomal, making up 1,371 out of 2,022 interolog pairs (see assignment method details in S1 Text). The 0–19% and 60–100% identity groups contain about half ribosomal and half non-ribosomal interolog pairs; however, the 19–34% and 34–60% groups contain a large majority of ribosomal pairs.

Given that π-stacking is the least abundant type of contact in our protein interaction dataset (see supplementary results in S1 Text), we decided to not further divide the π-stacking contact data into sequence identity or ribosomal/non-ribosomal groups, as these groups would contain very few data points. Henceforth, we will focus on atomic contacts, apolar contacts, H-bonds, and salt bridges.

Fig 4 and panel B in S3 Fig show that the conservation of H-bonds and salt bridges is especially low for groups of interologs with minimum interface sequence identity below 60%. Even in the groups of closest interologs (minimum interface sequence identity 60% to 100%), the average contact conservation is only 65% for H-bonds (61% for ribosomal interologs and 68% for non-ribosomal interologs) and 55% for salt bridges (same average value for ribosomal and non-ribosomal interologs). On the contrary, atomic and apolar contacts are well conserved even in pairs of remote interologs with minimum interface sequence identity below 19%; the average conservation in this group for atomic contacts is 56% (54% for ribosomal interologs and 58% for non-ribosomal interologs).

The contact conservation behavior is similar for ribosomal compared to non-ribosomal interologs, provided that we account for the different interface sequence identity composition of the two subsets (compare Fig 4A with Fig 4B). Most identity categories and contact types display a slightly higher conservation for non-ribosomal interfaces, compared to ribosomal interfaces (Table A in S2 Text). The difference is not always significant and cannot be obviously linked to a sequence identity bias in each sequence identity category (Table B in S2 Text). For example, a subgroup of the 19–34% ribosomal interologs displays higher than expected contact conservation, for all types of contacts. This subgroup corresponds to a number of highly similar Argonaute proteins interacting with very different RNA sequences in a structurally very similar manner (see below and panel B in S7 Fig). The difference between

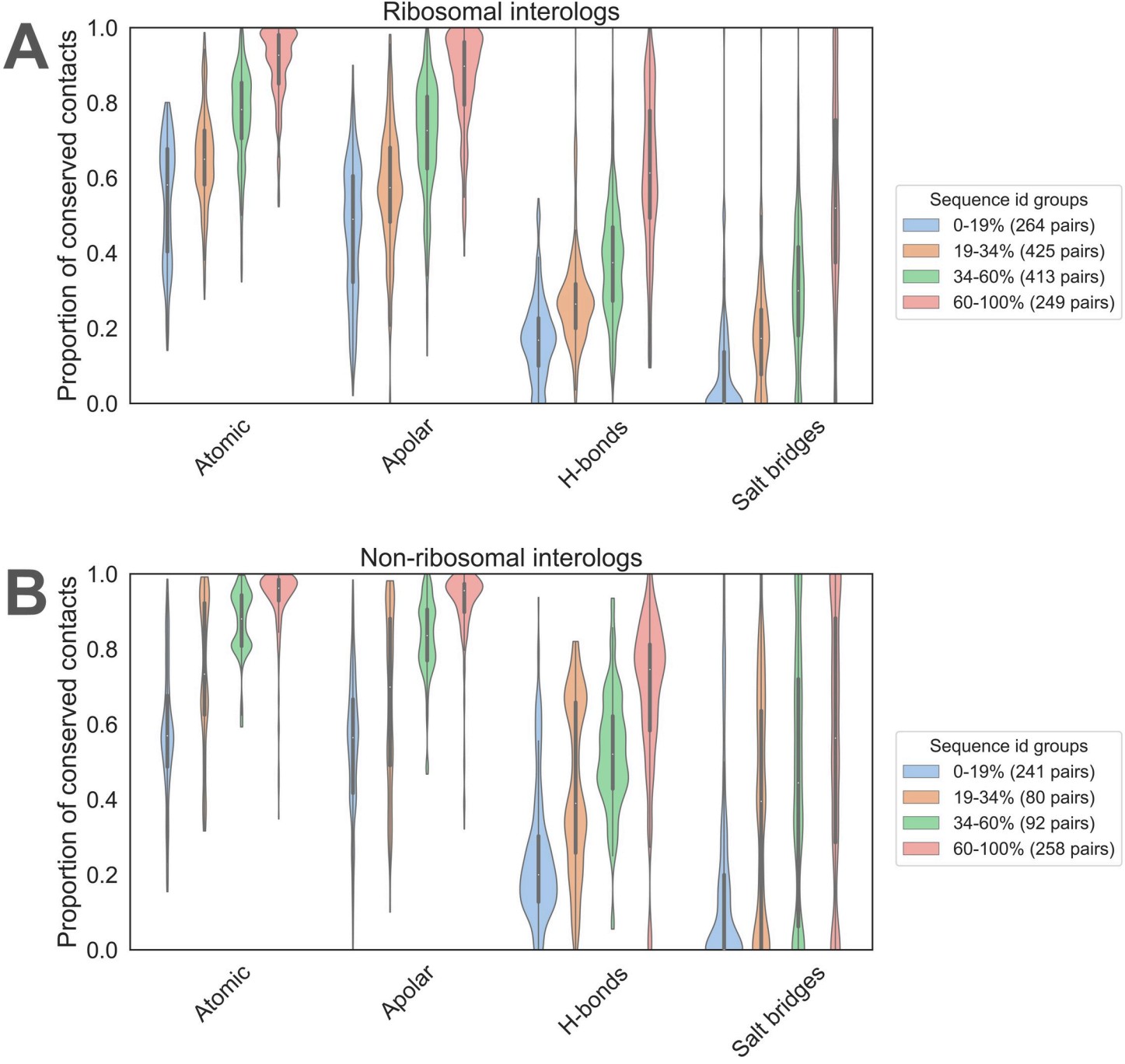

**Fig 4. Interface contact conservation results.** Violin plot distributions of contact conservation for distance-based atomic contacts (Atomic), apolar contacts (Apolar), H-bonds, and salt bridges, for (A) ribosomal and (B) non-ribosomal pairs of interologs separated into four groups of interface sequence identity (blue: 0–19%, brown: 19–34%, green: 34–60%, red: 60–100%). The legend indicates how many interolog pairs are included in each category.

ribosomal and non-ribosomal interologs disappears when shuffling or resampling the interface.

We verified that the observed trends were not due to the definition of our contact conservation metrics (see S5 Fig and supplementary results in S1 Text). We note that the more distant the interologs, the lower the interface overlap between them (panels A and B in S6 Fig). The

0–19% sequence identity group in particular has roughly one third fewer structurally aligned amino acid-nucleotide pairs despite similar interface sizes (see supplementary results in S1 Text and panels C and D in S6 Fig), hence contact conservation in this group might be moderately over-estimated compared to other groups because many amino acid-nucleotide pairs cannot be structurally aligned due to strong divergence. The challenges in structural alignment of such divergent interfaces will also translate into challenges for template-based modeling based on such remote templates. On the contrary, templates at 20% to 30% sequence identity do not seem to behave very differently from closer templates.

Contact conservation in the previously defined random baselines shows a small dependency on the sequence identity category of the interolog pair before shuffling/resampling (panels C and D in S4 Fig). This reflects an increasing correlation between CA-C3' distances of aligned amino acid/nucleotide pairs with increasing sequence identity (see supplementary results in S1 Text).

**Contact conservation within interolog groups.** Ribosomal or non-ribosomal interfaces can be further grouped according to structural homology connections; as mentioned earlier, this grouping is consistent with evolutionary relationships within protein families and RNA clans. We organized the 2,022 pairs of interologs into 141 groups, including 93 ribosomal and 48 non-ribosomal groups (see Methods). In each group, a node is an interface and interologs are connected by edges; the resulting graphs are not always complete, as some interfaces are not directly connected (not interologs) but connect through other interologs. We labeled each of these groups with the most represented ECOD and (whenever possible) Rfam domains among all interfaces in the group. To illustrate the variety of situations reflected by the global distribution of contact conservation, we examined a few of the largest interolog groups, which also correspond to well-known families of protein-RNA interfaces that are known to involve unique interaction properties [48] (see S7 Fig and supplementary results in S1 Text). One of the largest interolog groups involves proteins from the well-studied RRM (RNA recognition motif) family [48,49]. In our dataset of high-resolution interfaces, this group contains 29 interfaces connected through 133 interolog pairs, spread over the full range of interface sequence identities. In the RRM group, some pairs display very low H-bond conservation despite high sequence identity and high atomic contact conservation; a single interface structure (at 2.5 Å resolution) is responsible for this behavior, highlighting the importance of our strict selection criteria to perform in-depth analysis of H-bond conservation (panel A in S7 Fig). The second largest interolog group (19 interfaces connected through 154 interolog pairs) involves proteins with domains from the ribonuclease H-like family, including some Argonaute proteins, but also homologs such as endonuclease V. A number of pairs in this group actually display an outlier behavior in our analysis, as they have high values of contact conservation for atomic and apolar contacts as well as H-bonds and salt bridges, despite minimum interface sequence identity falling in the 19–34% or 34–60% identity group. These outlier pairs correspond to highly similar Argonaute proteins interacting with very different RNA sequences in a structurally very similar manner (panel B in S7 Fig). Other large interolog groups include well-studied interfaces between diverse ribosomal proteins (such as the L10e domain with bacterial large subunit ribosomal RNA, panel C in S7 Fig, or the signal recognition particle SRP family) and non-ribosomal families such as RNA helicases [50] (panel D in S7 Fig), viral RNA polymerases and Cas9 protein families. Analysis of each of these groups can bring interesting insights into the specifics of contact conservation for a given family of protein-RNA complexes. However, smaller groups are difficult to analyze because they contain very few data points. Importantly, each interolog group can be explored in the web interface and Jupyter notebook associated with the present study (see last Results section and Data Availability Statement).

**Interface determinants of atomic and apolar contact conservation.** We then explored how different local interface properties might influence the conservation of atomic and apolar contacts: secondary structure of interface amino acids and base pairing of interface nucleotides, interface subregions assigned depending on solvent accessibility, and evolutionary conservation derived from a protein multiple sequence alignment. We performed this analysis on the full dataset of interolog pairs, to identify global trends supported by a large number of data points. However, the tendencies reported below also hold when analyzing the ribosomal and non-ribosomal interologs separately.

We categorized contacts for each pair of interologs based on the type of secondary structure (helix, strand, or coil) in which amino acids and their structural equivalents are involved (see Methods). When amino acids have the same secondary structure type in both interologs, whether this secondary structure type is helix, strand or coil, the average proportion of conserved atomic contacts is 73%; in contrast, when amino acids change secondary structure, the average conservation drops to 55% (panel A in S8 Fig). A similar but even stronger trend is observed in the analysis based on RNA secondary structure, i.e. whether nucleotides are base-paired or not in one or both interologs. The average contact conservation ratio drops from 90% (respectively 68%) when nucleotides remain base-paired (respectively, unpaired) in both interologs to 23% when they change base-pairing status (panel B in S8 Fig). Strikingly, the tendency is the same within pairs of remote interologs with low interface sequence identity (panel C in S8 Fig). When nucleotides change base-pairing status between interologs, non-conserved contacts correspond to a vast majority of contacts made by a base-paired nucleotide, and lost in the interolog where the nucleotide is unpaired, rather than the reverse (see supplementary results in S1 Text). A possible explanation for these non-conserved contacts might thus be that the loss of nucleotide base-pairing leads to increased flexibility, compared to base-paired nucleotides that have more limited possible contacts with the protein.

To further investigate determinants of atomic contact conservation, we classified protein interface amino acids into core and rim regions (see Methods). Numerous studies have previously emphasized the distinctive characteristics of these subregions concerning composition and evolutionary properties in the context of protein-protein interactions [36,51,52]. We found that for protein-RNA interologs, atomic contacts within the core exhibit significantly higher conservation compared to contacts in the rim region (panel A in S9 Fig). Specifically, the average conservation of atomic contacts involving amino acids from the core regions in both interologs is 85%. In contrast, the average conservation of atomic contacts involving at least one residue from the rim region in any of the two interologs is 72 to 74% (p-value < 4e-56 based on Wilcoxon rank sum tests). Note that this analysis is limited to contacts involving residues that remain at the interface in both interologs, therefore removing a large fraction of non-conserved contacts (see next section), which leads to rather high average conservation percentages.

Finally, we analyzed atomic contact conservation according to position-specific amino acid evolutionary conservation derived from a protein multiple sequence alignment (see Methods). This property is especially interesting in the perspective of predictive developments, since it can be derived without any knowledge of the protein-RNA interface structure. We defined four groups of evolutionary conservation according to the minimum amino acid conservation within a pair of structurally aligned contacts: 0–30%, 30–50%, 50–70%, and 70–100%, for which the average atomic contact conservation is 47%, 53%, 58%, and 68%, respectively (panel B in S9 Fig). This result confirms that the most evolutionarily conserved amino acids within a protein family are also the positions that conserve most often their protein-RNA structural contacts.

### Analysis of non-conserved atomic contacts

**Non-conserved contacts involving residues that no longer make contacts.** Given the limitation to interface modeling that non-conserved contacts represent, we analyzed them in greater detail. We first examined the cases where either the amino acid or the nucleotide (or both) forming the non-conserved atomic contact not only "loses" the contact we are considering, but altogether does not make any contacts in the interolog; due to our contact-based definition of interface positions, this means that this amino acid or nucleotide no longer belongs to the interface. We previously identified this phenomenon, which we called "switching out of the interface", as a major scenario of non-conservation observed in protein-protein interface evolution [36].

In our dataset of 2,022 protein-RNA interolog pairs, on average, 11% of interface amino acids (weighted by the number of atomic contacts they are involved in) switch out of the interface in the interolog (Fig 5A) while the fraction is only 5% for interface nucleotides (S10 Fig). These fractions rise to 21% and 17%, respectively, when not weighted by the number of atomic contacts. Despite the low proportions of switching out residues, we found that switching out was involved in a large proportion of non-conserved atomic contacts, 47% on average. Strikingly, this proportion does not vary very strongly with sequence identity (Fig 5B). This means that in remote interologs, the fraction of non-conserved contacts is higher compared to close interologs, but so is the fraction of switching out residues, and the proportion of non-conserved contacts related to switching out does not vary much. Among non-conserved contacts related to switching out, 61% are cases where the amino acid switches out of the interface, 29% where the nucleotide switches out, and 10% where both switch out.

Because switching out affects a small fraction of interface residues but a large fraction of the non-conserved contacts, identifying switching out residues could be interesting in a prediction perspective, before considering the conservation of each interface contact. Switching out residues display characteristic features that would be useful in this regard; for instance, 26% of (unweighted) amino acids from the rim interface region switch out, as opposed to 17% from the core region.

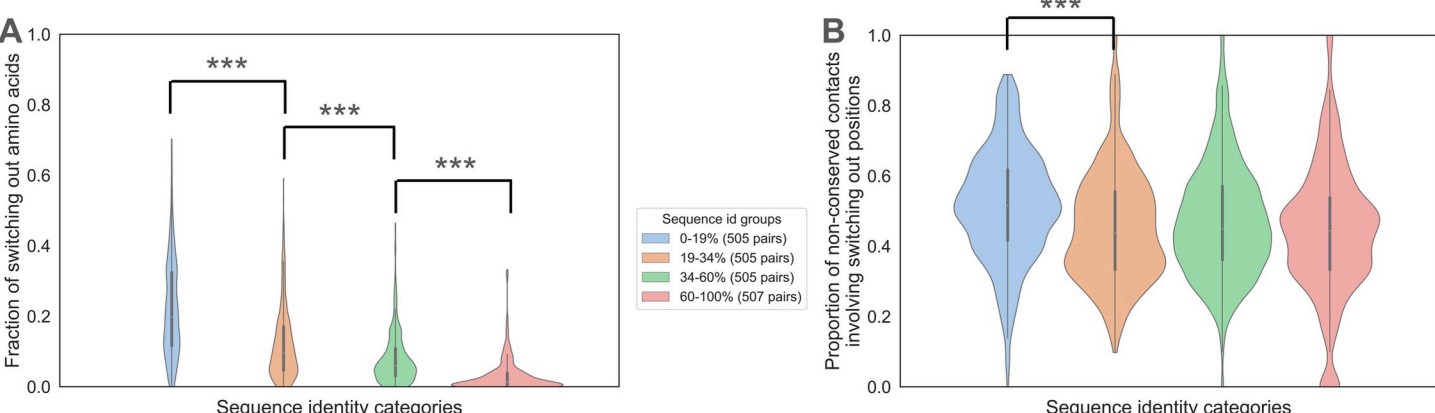

**Fig 5. Analysis of switching out in contact non-conservation.** A: Violin plot distribution of the percentage of switching out amino acids (weighted by the number of atomic contacts in which each amino acid is involved) across the four ranges of interface sequence identity. The differences between any two distributions of switching out among the four groups of sequence identities are statistically significant (p-value < 1.6e-13 in Wilcoxon rank sum tests). B: Violin plot distribution of the unweighted percentage of non-conserved contacts related to switching out across the four ranges of interface sequence identity. The difference between 0–19% and 19–34% distributions is significant (p-value = 8.4e-13 in Wilcoxon rank sum test), while the differences between 19–34% and 34–60%, and 34–60% and 60–100% are not (p-values = 0.035 and 0.015 in Wilcoxon rank sum tests).

**Compensation scenarios for non-conserved H-bonds.**    Numerous studies show that hydrogen bonds play a crucial role in protein-RNA interactions by contributing to the specificity, stability and function of the complexes [40,42,53,54]. As we observed a marked versatility of hydrogen bonds, even between interologs with high interface sequence identity (Fig 4), we conducted an in-depth analysis of non-conserved H-bonds, aiming to investigate potential recovery mechanisms. In this analysis, we focused only on H-bonds involving the side chains of amino acids; this simplifies the interpretation of possible compensation scenarios involving evolutionary mutations affecting the amino acid nature. This explains why the fraction of conserved H-bonds (light green slices in Fig 6A) is different compared to S3B Fig.

Across the four groups of interface sequence identity, we accounted for a large part of the non-conserved H-bonds due to switching out or lost polarity of the sidechain (light blue and dark green slices in Fig 6A, illustrated by light blue and dark green boxes in Fig 6B). A large portion of the remaining non-conserved H-bonds (where the residues remain at the interface and the amino acid retains a polar sidechain) can be explained by recovery mechanisms involving other intermolecular H-bonds with a non-structurally aligned position in the

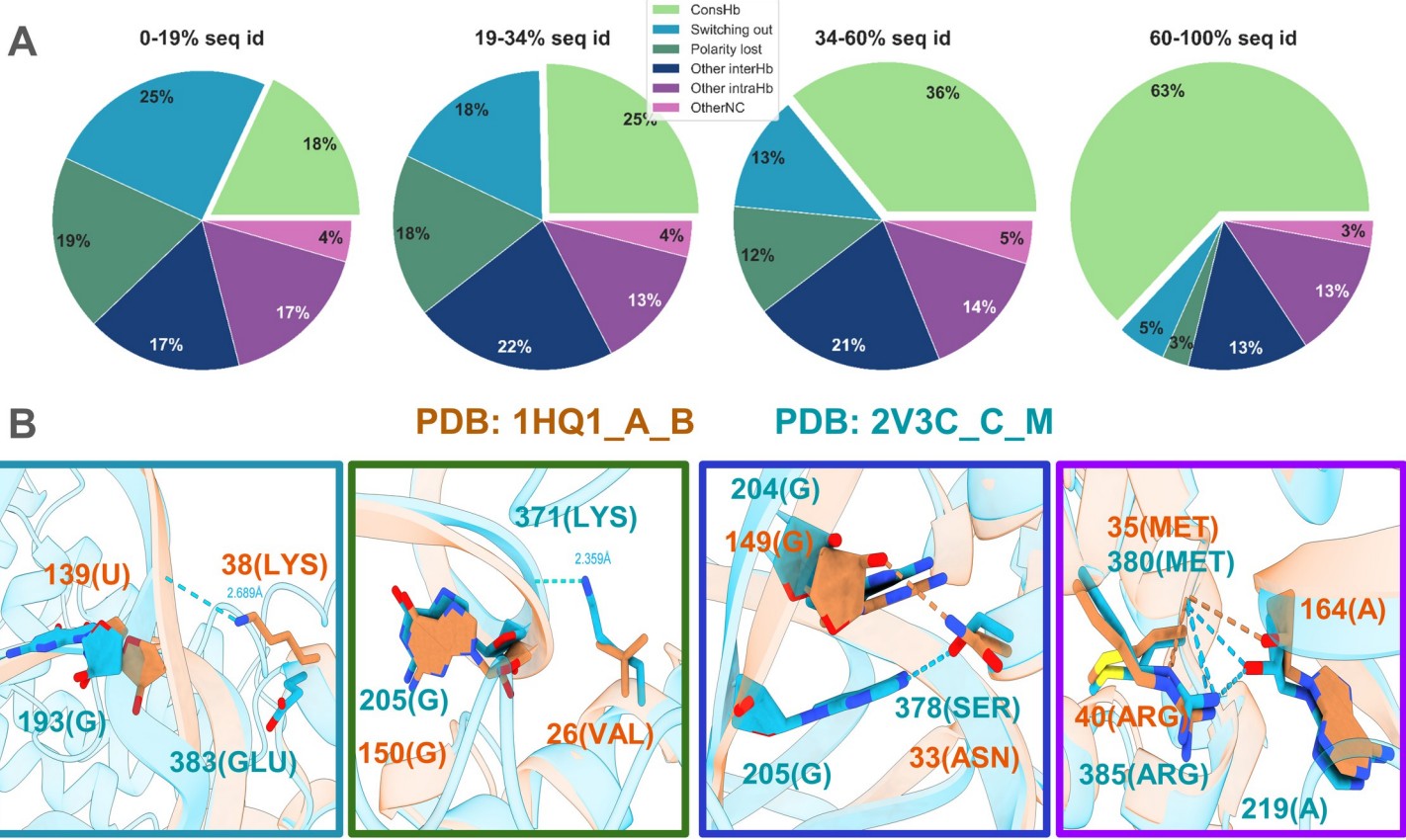

**Fig 6. H-bond conservation and recovery mechanisms of non-conserved hydrogen bonds.** (A) Pie plots representing the scenarios of H-bond conservation across the four groups of interface sequence identity. The six categories represent conserved hydrogen bonds (ConsHb), non-conserved H-bonds due to switching out (Switching out), non-conserved H-bonds due to non-polar character of the interface amino acid, leading to a loss of their ability to form H-bonds through their sidechain (Polarity lost), non-conserved H-bonds involving amino acids that retained the ability to form H-bonds with neighboring nucleotides (Other interHb), non-conserved H-bonds involving amino acids that retained the ability to form H-bonds but did not form them with nucleotides, rather with adjacent amino acids (Other intraHb) and other non-conserved H-bonds where the amino acid retained the ability to form H-bonds but did not form them either inter- or intra-molecularly (OtherNC). (B) Structural illustrations of four categories from the pie plot, from left to right: Switching out (switching out of nucleotide 193(G)), Polarity Lost (26(VAL) lost the ability to form H-bonds through its sidechain), Other interHB (378(SER) forms a hydrogen bond with nucleotide 205(G), not structurally aligned with 149(G)) and Other intraHB (40 (ARG) no longer forms a H-bond with a nucleotide but forms an intramolecular H-bond with amino acid 35(MET)).

interolog (dark blue slices in Fig 6A, illustrated by dark blue box in Fig 6B) or intramolecular H-bonds with a neighboring amino acid (dark purple slices in Fig 6A, illustrated by dark purple box in Fig 6B). A small fraction of the non-conserved H-bonds remains unexplained (purple slices in Fig 6A) and may correspond, for instance, to larger distance contacts mediated by water molecules. These findings provide guidelines for both template-based modeling and assessment of interfaces in protein-RNA docking poses.

**Compensation scenarios for non-conserved salt bridges.** Salt bridges form a subset of hydrogen bonds involving short-distance ionic interactions. The conservation of salt bridges among protein-RNA interologs is even lower than for H-bonds (Fig 4). We investigated recovery scenarios in a manner very similar to H-bonds. Similarly to H-bonds, a large part of the non-conserved salt bridges can be explained by switching out or loss of the basic chemical property of the amino acid (light blue and dark green slices in Fig 7A, illustrated by light blue and dark green circles in Fig 7B). However, among the remaining non-conserved salt bridges, only a fraction can be explained by recovery mechanisms involving other intermolecular salt bridges with a non-structurally aligned position in the interolog (dark blue slices in Fig 7A) or

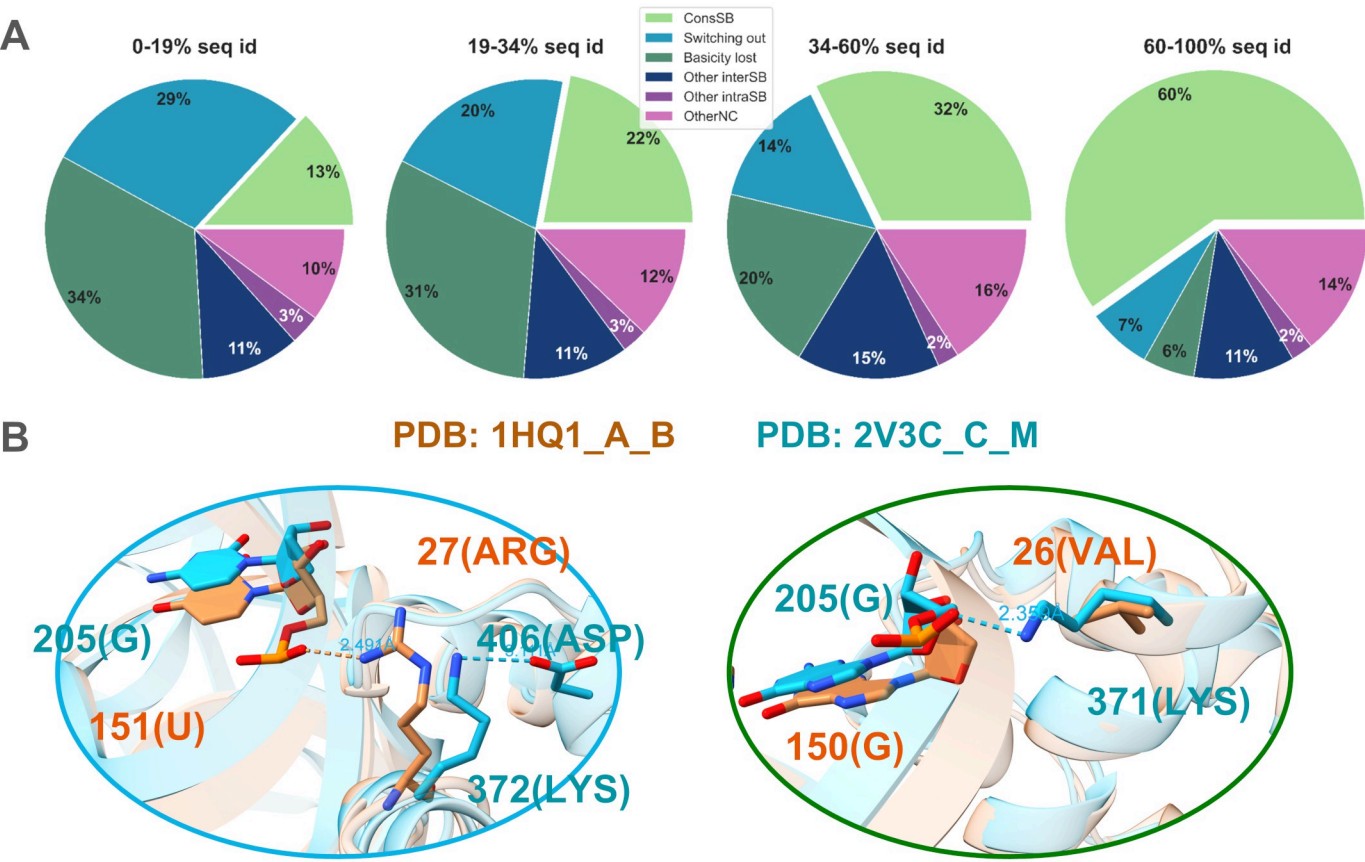

**Fig 7. Salt bridge conservation and recovery mechanisms of non-conserved salt bridges.** (A) Pie plots representing the scenarios of salt bridge conservation across the four groups of interface sequence identity. The six categories represent conserved salt bridges (ConsSB), non-conserved salt bridges due to switching out (Switching out), non-conserved salt bridges because the interface amino acid is not basic in the interolog (Basicity lost), non-conserved salt bridges involving basic amino acids that form salt bridges with neighboring nucleotides (Other interSB), non-conserved salt bridges involving basic amino acids that form salt bridges with neighboring acidic amino acids (Other intraSB), and other non-conserved salt bridges (OtherNC). (B) Illustrations of non-conserved salt bridges due to switching out (left, light blue circle) and loss of basic character (right, dark green circle). In the left panel, the salt bridge between 27(ARG) and 151(U) in interface 1HQ1_A_B is not conserved due to switching out of amino acid 372(LYS) from interface 2V3C_C_M; this amino acid still forms an intramolecular salt bridge with neighboring amino-acid 406(ASP). In the right panel, salt bridge between 371(LYS) and 205(G) in interface 2V3C_C_M is not conserved in 1HQ1_A_B due to hydrophobicity of the 371(LYS) structural equivalent, 26(VAL).

intramolecular salt bridges (dark purple slices in Fig 7A). This leaves a fraction of unexplained non-conserved salt bridges (light purple slices in Fig 7A) that is larger compared to H-bonds. Amino acids which remain basic but no longer form salt bridges can instead form intermolecular H-bonds with neighboring nucleotides, but this scenario accounts for a very small fraction (1–2%) of the total (hence, not singled out in Fig 7 and included in the OtherNC category). These amino acids could also form longer distance electrostatic interactions or interactions mediated by solvent molecules.

## Web interface for the exploration of interolog groups and contact conservation

To complement our study, we have developed a user-friendly web interface (see Methods) allowing for the dynamic exploration of groups and pairs of interologs and the interactive structural visualization of the 765 protein-RNA interfaces and 2,022 pairs of homologous interfaces in our dataset. Starting from a global view of interolog groups (Fig 8A, also displayed on the web interface home page), users can click on a group to display its network representation (Fig 8B) and details about the interolog pairs in this group. Users can then proceed to explore either a single interface (by clicking on a node of the network) or a pair of interologs (by clicking on an edge of the network). In the case of a single interface, contact details are displayed and the structure of the interface can be explored interactively (Fig 8C), including contacts involving any given amino acid or nucleotide; various modes of representation can be chosen. In the case of a pair of interologs, we display precomputed information about the conserved and non-conserved contacts and the aligned interface structures can also be explored interactively (Fig 8D). On all three pages accessible from the navigation menu of the web interface (home page, interolog group page, and interface list page), users can search for specific keywords within PDB, ECOD and Rfam descriptions of the macromolecular components, to allow for the extraction of biologically relevant information from our data. Users can also download relevant information as tables for either the full datasets or a given interface or pair of interologs.

This web interface is freely accessible at https://bioi2.i2bc.paris-saclay.fr/django/rnaprotdb/.

## Discussion

Protein-RNA interactions are critical for many biological processes, including gene expression, RNA processing, and translation. Understanding the mechanisms underlying these interactions is pivotal for improving structural prediction methods and for further drug design targeting these interactions. Our in-depth analysis of structural interologs provides crucial insights into the evolution of protein-RNA interactions. We first highlighted that when protein-RNA interfaces share a minimum of 25% sequence identity, they share similar overall interface structures provided they display a similar protein fold. This is especially important for structural modeling of protein-RNA complexes, when aiming to identify templates or to retrieve relevant homologous sequence data. We then built a large dataset of over 2,000 pairs of well-resolved protein-RNA structural interologs, which enabled us to study the conservation of protein-RNA interface structures in great atomic detail. The high conservation of distance-based atomic contacts, and especially of apolar contacts involving base-paired nucleic acids, offers a reliable foundation for transferring these structural features of protein-RNA complexes in template-based modeling. In parallel, the conservation of atomic and apolar contacts suggest that they are key for interface stability and underscores their potential as robust features for the development of machine learning predictors and for better identification of correct poses in template-free docking (e.g. through propensity-based scoring functions taking evolutionary

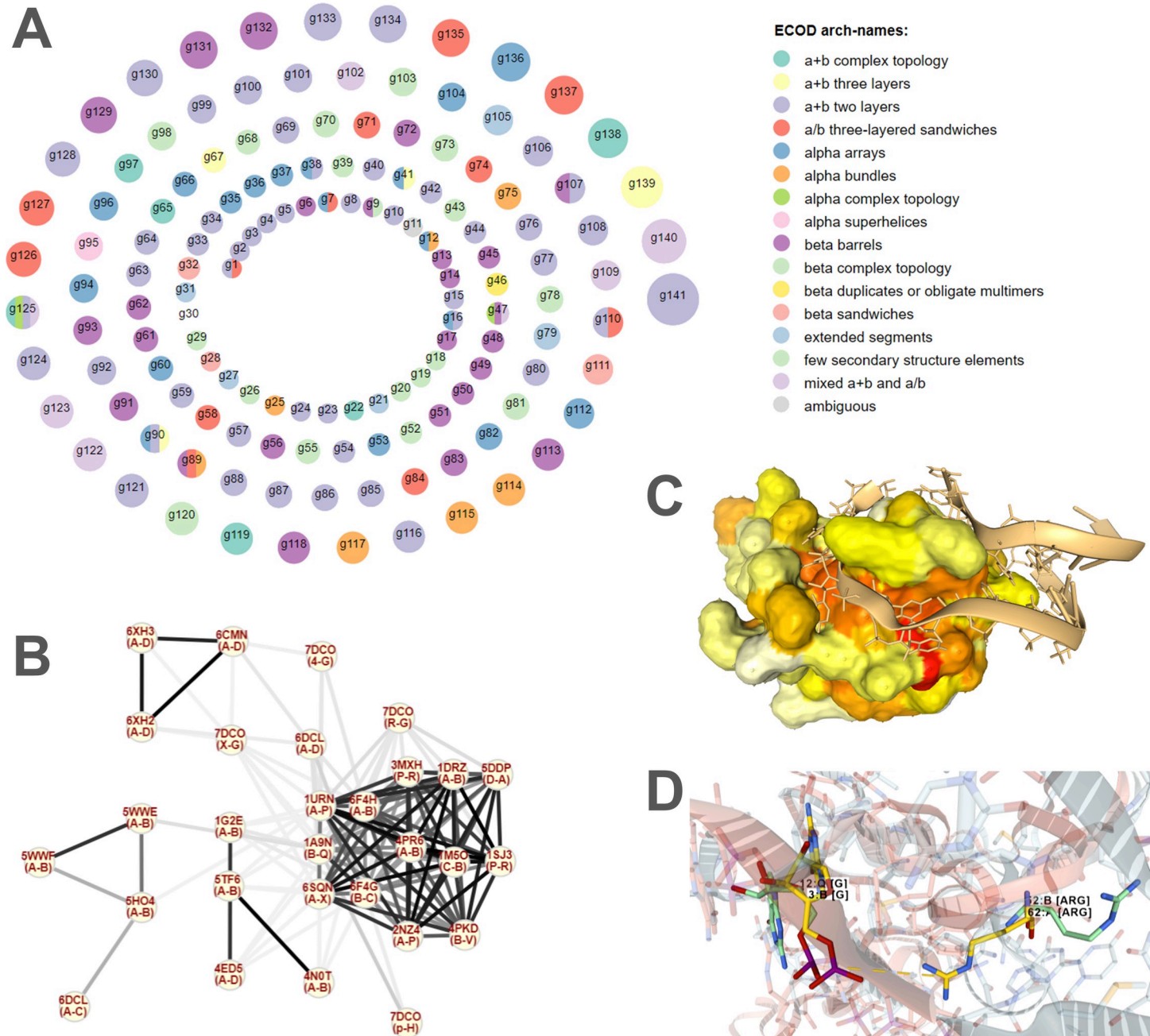

**Fig 8. Exploration of protein-RNA interfaces and interologs in our datasets.** (A) Overall view of the 141 interolog groups. Nodes are colored by ECOD architecture level information from the common ECOD domains in each group. (B) Network view of the largest interolog group (g141). Nodes are interfaces and edges are homology relationships between interfaces. Edge colors reflect interface sequence identity (gray color scale—the darker, the higher the identity). (C) 3D structural visualization of a single interface (1A9N_B_Q). The illustration shows a surface view for the protein colored by evolutionary conservation (from white to red, most variable to most conserved) and a cartoon view for the RNA. Interface residues are highlighted as licorice. (D) 3D structural comparison of two homologous interfaces. The visualization is focused on ARG 62 of 5WWE_A (yellow sticks). The structurally aligned position is shown (here, ARG 52 in 1A9N_B, green sticks) as well as the corresponding contacting nucleotides (G 3 in 5WWE_B, aligned with G 12 in 1A9N_Q). These views can be explored interactively in the web interface of RNAprotDB (from panel A to panel B to panels C and D): https://bioi2.i2bc.paris-saclay.fr/django/rnaprotdb/.

information into account, a strategy we previously found successful for protein-protein inter-actions [37]). We further investigated the importance of interface sub-region and secondary structure conservation. We evidenced increased contact conservation for evolutionarily more

conserved interface regions, meaning that the most conserved interface amino acids also have the most contact conservation. This latter feature is especially interesting in a prediction context, e.g. when wondering which contacts can be transferred from a known protein-RNA interface to a target homologous complex, since computing evolutionary conservation and mapping it onto the protein surface does not require prior experimental knowledge about the target protein-RNA interface structure. Conversely, the versatility of H-bonds and salt bridges, even between interfaces with high sequence identity, prompts cautious consideration in template-based modeling, and may offer insights into modulating the specificity of protein-RNA complexes, especially by considering the possible recovery mechanisms involving neighboring nucleotides or amino acids. Additionally, our analysis and the resources we provide cover a diverse range of protein and RNA families. For the largest analyzed interolog groups, our carefully crafted dataset and our insights on protein-RNA interface evolution open new possibilities for tailoring family-specific scoring metrics to further improve protein-RNA specificity predictions. As an illustration of such specificity studies, a recent study used curated sequence and structure data to gain insights into the recognition code for RRM-RNA interactions and derive a specificity prediction score for this family [49].

In a previous study, we analyzed a dataset of around 1,000 pairs of protein-protein structural interologs [36], which provides us with a basis for comparing protein-protein and protein-RNA interface evolution. Protein-RNA interface evolution shows distinctive features, with protein-RNA interfaces displaying a lower proportion of non-conserved contacts for all contact types, even though our protein-RNA interface dataset contains many remote interologs. The proportion of non-conserved contacts explained by switching out is slightly higher in protein-RNA interologs compared to protein-protein interologs, despite the proportion of switching out residues being much lower. Altogether, this would suggest that protein-RNA interfaces are less versatile than protein-protein interactions, even if previous studies show greater flexibility for both protein and RNA in protein-RNA interfaces compared to protein-protein interfaces [19]. These observations help us to identify the most relevant keys to guide the modeling of protein-RNA interactions.

Other possible implications of protein-RNA interface evolution include insights into prebiotic world interactions. Those were previously investigated through the analysis of small datasets of protein-RNA complexes resulting from in vitro evolution [55,56]. However, in the present study we rather perform a statistically robust analysis of a large number of homologous interfaces to derive general principles of protein-RNA interface evolution.

In conclusion, our study offers insights to advance understanding of protein-RNA interactions and their evolution. It lays the groundwork for developing propensity-based scoring functions and refining structural prediction methods. Finally, it opens perspectives into protein-RNA interface modulation and possible design of drugs targeting protein-RNA interfaces. Future research should leverage these evolutionary insights in advanced protein-RNA modeling approaches building upon the latest deep-learning advances, such as AlphaFold3 [35] and RosettaFold2NA [33]. Our insights into conservation of protein-RNA interfaces suggest that these methods may be leveraging structural information from remote interologs in the training dataset, raising the as yet unanswered question of how to fairly assess their generalizability.

## Methods

### Interface database construction

Fig 1 depicts the pipeline used to obtain the interface dataset. We collected 3D atomic structures of protein-RNA complexes from the Protein Data Bank [9] that contained at least one protein polymer entity and one RNA polymer entity, resulting in a total of 4,173 structures as

of 21 February 2022. For each PDB entry, we used Gemmi [57] to generate coordinates of the biological assembly, which better represents the complexes in their functional form compared to the asymmetric unit, and to assess pairwise heavy-atom distances between each amino acid and each nucleotide. We defined a contact between an amino acid and a nucleotide if the minimum heavy-atom distance between them was less than 5Å. The interface is defined as a set of all amino acids and nucleotides involved in such inter-chain contacts. We also processed all generated biological assembly CIF files using the software tool x3DNA v2.4 to assign base pairs in RNA structures [58,59]. The complexes were divided into binary interfaces consisting of one protein chain contacting either one RNA chain or two base-paired RNA chains if at least one amino acid from the protein chain is in contact with two base-paired nucleotides, one from each RNA chain. A total of 114,965 binary interfaces were identified in the complete interface dataset.

As we aimed at a high-resolution study of interface contact conservation, we filtered out NMR structures and structures with a resolution worse than 2.5 Å, as well as protein chains shorter than 30 amino acids or containing only CA atoms, and RNA chains shorter than 10 nucleotides or containing only P atoms. We also excluded interfaces with fewer than 5 protein or 5 RNA interface residues. Clustering was performed on the resulting 3,419 protein-RNA (binary or ternary) interfaces, to group strictly redundant interfaces containing protein chains with 100% sequence identity and RNA chains with 99% sequence identity or more (the RNA redundancy threshold was taken from [12]). We used MMseqs2 [60] and CD-HIT [61] for clustering protein and RNA sequences, respectively. The resulting interface dataset contains a representative (chosen as the interface with the best resolution) for each of the 977 interface clusters.

## Interface analysis

We then generated detailed structural information for the dataset of 977 representative protein-RNA interfaces. Using Gemmi [57], we recorded the number of interacting atomic pairs (at a minimum distance of 5Å) involved in each amino acid-nucleotide contact and the number of apolar atomic pairs (focusing on carbon atoms). To characterize contacts of different natures [62,63], we used x3DNA v2.4 [58,59] to identify π-stacking interactions, salt bridges and hydrogen bonds (H-bonds), including details about whether each H-bond involves side-chain/backbone for the amino acid and sugar/base/phosphate for the nucleotide. We also assigned secondary structures for the protein amino acids using the DSSP algorithm [64] through the Biopython Bio.PDB module [65]. We converted the DSSP output into three classes: helix (H, including DSSP H, G, and I categories), strand (E, including DSSP B and E categories) and coil (C, including DSSP T, S and - categories).

To determine the evolutionary divergence of each protein chain in our dataset, we generated multiple sequence alignments (MSAs) using 1 iteration of HHblits version 3.0.0 [66] against the Uniref30 [67] version of February 2022. We filtered the obtained MSAs with HHfilter [66] using a 30% minimum sequence identity with the query sequence and a diff parameter to 80, limiting the number of sequences while ensuring sequence diversity in the MSA. The Rate4Site software package was then used to calculate a conservation score for each position in the protein sequence [68,69]. The calculated values were rescaled from 0 to 100, with higher values associated with more conserved residues likely indicative of functional importance. Rate4Site calculations failed for a subset of 38 interfaces in our dataset, and the corresponding interologs are therefore excluded from evolutionary conservation analysis.

We divided the protein interface according to the core-rim model [51]. We calculated the relative accessible surface area of the complex (rASAc) for the amino acids in each interface using the Python module of freesasa, using a probe radius of 1.4 Å. Interface rim protein

residues are those that have rASAc > 25% and all other protein interface residues were assigned to the interface core region following previous work [51,52].

## Structural alignment

We generated coordinates in the PDB file format for each of the 977 representative binary interfaces. Using PDBFixer from the openMM Python module [70], we converted non-standard amino acids/nucleic acids to their standard counterparts (using the Chemical Component Dictionary from the PDB) and we added missing atoms. We then performed all-against-structural comparisons between the 977 interfaces, excluding pairs of interfaces that belong to the same PDB entry and have one chain in common. For the remaining approx. 445,000 pairs of interfaces, we used TM-align (Version 20190822) for protein structural alignment [43], RNA-align (Version 20191021) for RNA structural alignment [44], and the updated version of MM-align (Version 20191021) for protein-RNA interface structural alignment [45]. For each pair of compared interfaces, we used the protein, RNA, and interface TM-scores provided by each software tool (respectively TM-align, RNA-align and MM-align). The software provides TM-scores normalized by each molecule/interface separately; we retained only the TM-scores normalized by the smallest molecule/interface, as many of our comparisons involve complexes of very different sizes.

## Structural comparison metrics

We converted the alignment file from MM-align into a dictionary of structural correspondence listing structurally aligned pairs of amino acids and nucleotides from each pair of interfaces. The interface overlap was computed for protein and RNA as the number of interface amino acids or nucleotides that had a structural correspondent normalized by the size of the smallest (protein or RNA) interface. For any given pair of structurally compared interfaces, all following analyses of contact conservation were restricted to amino acids and nucleotides with structural correspondents in both interfaces.

After superimposing interfaces with MM-align, we calculated the interface RMSD (Root Mean Square Deviation), which measures the structural similarity between two protein-RNA complexes at their interface, using coordinates of the P atoms for interface nucleotides and CA atoms for interface amino acids.

In this study, we are using structural alignment as a gold standard since all considered interfaces have good resolution structural coordinates. Structure-based sequence identity was computed for two aligned protein chains or RNA chains based on structural interface alignment results, as the number of identical positions divided by the number of aligned positions. Interface sequence identity was computed by considering only interface positions (as defined by the minimum heavy-atom 5Å distance criterion). In general, e.g. with the goal of interface template-based modeling or MSA-based predictions, interface structures might not be known in advance. Hence, as a reference, we also computed a sequence-based sequence identity by extracting reference sequences from the PDB (canonical sequences, where non-standard amino acids and nucleotides are converted to their standard counterpart as much as possible) and aligning separately protein sequences and RNA sequences for each pair of compared interfaces, using the FASTA36 program [71] as was done in the previous study of template-based protein-RNA interface modeling [12]. Because this can lead to very short aligned segments for dissimilar molecules, we weighted the sequence-based sequence identity by the sequence-based alignment coverage (number of aligned positions divided by the length of the largest molecule).

### Evolutionary classification of RNA-binding domains and RNA chains

We organized the 2,022 pairs of interologs into groups. Using the networkx [72] Python package, we defined a graph where each of the 765 interfaces is a node and we added edges corresponding to pairs of structural interologs. Groups of interologs were defined as the connected components of this interolog graph. We obtained 141 groups containing between 2 and 29 interfaces.

ECOD (Evolutionary Classification of Protein Domains) [46] version 20230309 (develop288) was used to annotate protein chains with its hierarchical classification system based on evolutionary relationships and structural similarities [46,73]. We annotated each protein chain involved in a binary protein-RNA interface according to all ECOD domains containing amino acids within 5 Å of the RNA chain. Interfaces with more than one RNA chain were labeled with the union of all ECOD domains including residues from all pairwise protein-RNA interfaces. For each pair of interologs, we checked whether the sets of ECOD domain labels for the two protein chains (when available) were identical or overlapping. We used the ECOD T-group level for comparison (groups of homologs with similar topological connections). Each group of interologs was labeled with the most represented (or set of most represented) ECOD domains among all interfaces in the group; if no ECOD annotation was available or ECOD labeling was not unambiguously possible, then the group was labeled ambiguous.

Rfam [47] is an evolutionary classification of RNA families. Rfam families are grouped into clans when they are remotely homologous or when they can be aligned, but have distinct functions. We retrieved the mapping of Rfam (version 14.8) families and clans to all RNA chains belonging to the 765 interfaces in our dataset. We annotated pairs and groups of interologs with Rfam labels in the same manner as described for ECOD labeling.

### Calculation of interface contact conservation

The analysis of protein-RNA interface contact conservation was conducted similarly to our previous study of protein-protein interface evolution [36]. Interface contact conservation between a pair of structural interologs (homologous interfaces) was computed only for contacts involving amino acids and nucleotides with structural correspondents in both interfaces. The conservation of each type of contact was calculated using the Jaccard index (similarity coefficient), which is the ratio of the number of conserved contacts (formed in both homologous interfaces) to the total number of contacts formed in at least one interface. For atomic and apolar contacts, this ratio was weighted by the average number of atomic contacts between two positions (only for C atoms in the case of apolar contacts) if the contact existed in both interologs. If it only existed in one interolog, the ratio was weighted by the number of atomic contacts in the interolog where it existed. We also computed the conservation of H-bonds, salt bridges and π-stacking contacts assigned by x3DNA v2.4 by using the Jaccard index (no weighting).

For each pair of interologs, we computed the fraction of amino acids (respectively, nucleotides) switching out of the interface, by weighing each amino acid (respectively nucleotide) by the sum of all atomic contacts it makes within its respective interface. For each pair of interologs, the fraction of non-conserved contacts linked to switching out was computed as the unweighted ratio of the number of non-conserved contacts where the amino acid and/or the nucleotide switch out of the interface compared to all non-conserved contacts.

For all structural visualization aspects in this manuscript (apart from Fig 8), we used the ChimeraX software [74].

### Web interface for interolog exploration and visualization

The web server hosted at https://bioi2.paris-saclay.fr/django/rnaprotdb/ provides an interactive and comparative view of all 2,022 pairs of interologs in our database. It was generated

using Django version 4.1.7 [75] and uses NGL Viewer [76,77] to allow users to freely manipulate and explore the interolog structures and their pre-calculated features. Additionally, users can interactively explore interolog group networks thanks to the Cytoscape.js plugin version 3.28.1 [78]. The representations in Fig 8 were generated using these same tools.

## Supporting information

**S1 Fig. Histogram representing the number of interfaces per PDB identifier in the initial dataset of 4173 PDB identifiers.**
(PDF)

**S2 Fig. Supplementary information about the construction of the dataset of interologs.**
(PDF)

**S3 Fig. Supplementary interface contact conservation results.**
(PDF)

**S4 Fig. Interface contact conservation results for the two random baselines with either shuffled (panels A, C) or resampled (panels B, D) interfaces.**
(PDF)

**S5 Fig. Distributions of contact conservation using alternative metrics.**
(PDF)

**S6 Fig. Interface overlap depending on sequence identity category for ribosomal (panels A, C) and non-ribosomal (panels B, D) interologs.**
(PDF)

**S7 Fig. Distributions of atomic contact conservation (left plots) and H-bond conservation (right plots) depending on interface sequence identity for four interolog groups.**
(PDF)

**S8 Fig. Violin plots of atomic contact conservation depending on residue secondary structure properties.**
(PDF)

**S9 Fig. Violin plots of atomic contact conservation depending on amino acid properties.**
(PDF)

**S10 Fig. Supplementary analysis of switching out in contact non-conservation.**
(PDF)

**S1 Text. Supplementary methods and results.**
(PDF)

**S2 Text. Supplementary tables.**
(PDF)

## Acknowledgments

We thank R. Guerois and D. J. Zea for fruitful discussions and feedback on the manuscript. We thank E. Frezza and A. Taly for early feedback on the project. We thank M. Wojdyr for help with Gemmi and R. D. Schaeffer for help with ECOD. The initial code for the web interface to visualize interolog structures was inspired from the visualization interface developed by J. Rey together with P. Tufféry for the InterEvDock3 web server.

## Author Contributions

**Conceptualization:** Jessica Andreani.

**Data curation:** Ikram Mahmoudi, Chloé Quignot, Carla Martins, Jessica Andreani.

**Formal analysis:** Ikram Mahmoudi, Chloé Quignot, Carla Martins, Jessica Andreani.

**Funding acquisition:** Jessica Andreani.

**Investigation:** Ikram Mahmoudi, Chloé Quignot, Jessica Andreani.

**Methodology:** Jessica Andreani.

**Project administration:** Jessica Andreani.

**Software:** Ikram Mahmoudi, Chloé Quignot, Carla Martins, Jessica Andreani.

**Supervision:** Jessica Andreani.

**Visualization:** Ikram Mahmoudi, Chloé Quignot, Jessica Andreani.

**Writing – original draft:** Ikram Mahmoudi, Jessica Andreani.

**Writing – review & editing:** Ikram Mahmoudi, Chloé Quignot, Jessica Andreani.

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
