## [Decision Letter · Decision Letter 0]

22 Jul 2024

Dear Dr Andreani,

Thank you very much for submitting your manuscript "Structural comparison of protein-RNA homologous interfaces reveals widespread overall conservation contrasted with versatility in polar contacts" for consideration at PLOS Computational Biology.

As with all papers reviewed by the journal, your manuscript was reviewed by members of the editorial board and by several independent reviewers. In light of the reviews (below this email), we would like to invite the resubmission of a significantly-revised version that takes into account the reviewers' comments.

We cannot make any decision about publication until we have seen the revised manuscript and your response to the reviewers' comments. Your revised manuscript is also likely to be sent to reviewers for further evaluation.

Sincerely,

Rachel Kolodny

Academic Editor

PLOS Computational Biology

Arne Elofsson

Section Editor

PLOS Computational Biology

Reviewer's Responses to Questions

**Comments to the Authors:**

Reviewer #1: The paper addresses an important problem of protein-RNA interactions from structural, physicochemical and evolutionary perspective. The protein-RNA interactions play a key role in cellular mechanisms. Our ability to understand, model and manipulate these interactions to reveal biological mechanisms and for drug design is critically dependent on systematic studies of protein-RNA interfaces. Currently, such studies and the related modeling approaches significantly lag behind those focusing on protein-protein interactions. Advances in deep learning applications, such as AlphaFold3, do not negate but rather re-emphasize the need for protein-RNA knowledge resources. The paper in a significant way contribute to filling this knowledge gap. The authors provide a systematic analysis of the existing data on protein-RNA interactions, with great attention to details and interpretation of the results. The authors are experts on the subject. The paper is well written and illustrated. The related public resources will be of great use for protein-RNA studies. As such, the paper is of significant interest to the research community.

Reviewer #2: Mahmoudi et al present an analysis of the structural conservation of protein-RNA complexes across the PDB. To do so they first derive a dataset of 977 non-redundant protein-RNA complexes (representative interfaces), and clustered 765 of these into 2022 (interolog) pairs of homologous protein-RNA complexes. They show that apolar contacts are relatively well conserved across interologs, while H-bridges, salt bridges and pi-stacking contacts are poorly conserved, sometimes even across interologs with high sequence identity. They also show that the non-conservation of atomic contacts between interologs is often driven by what they call "switching-out" of the interface, mainly of the amino-acid making the contact, or by the nucleotide. The work is an interesting and valuable description/analysis of protein-RNA interactions.

The manuscript is generally well written and presented, although the clarity of some aspects could be improved (detailed below). Certain aspects of the analysis could also be improved/added, in particular regarding the contribution of ribosomal protein-RNA complexes in the dataset. This is a major point as it is now unclear whether this work describes mostly trends related to ribosomal protein-RNA complexes, or whether it describes protein-RNA complexes in general. The results on the conservation of certain contact types could also be improved by providing an expected “baseline”.

General comments

- I don't understand how the interolog’s structure shown in Fig 2C passed the first filtering steps. The TM-score for the two proteins is 0.74 because normalized by the smallest structure, but the coverage appears very low (the two RNA structures don't appear to bind on the same surface site, so how could it pass the initial thresholds?) - I don't understand how it passed the filtering step "MM-align succeeds in aligning protein and RNA in interface", nor the filtering step on interface TM-score > 0.5.

- The previous comment involves the calculation of an “interface TM-score” but I could not find a description in the methods. This should be described in detail.

- The authors wrote: "Distance-based contacts are also much more conserved compared to what we had found in a previous study of homologous protein-protein interfaces. In contrast, hydrogen bonds, salt bridges, and π-stacking interactions are very versatile in pairs of protein-RNA interologs, even for close homologs with high interface sequence identity." An important conclusion of the work is that certain contact types are more conserved than others, in particular apolar contacts appear more conserved than others. At the same time, because apolar contacts are much more frequent to begin with, a random reshuffling of contacts will naturally result in apolar contact being frequent and others being rare. For that reason, this observation should be compared to a carefully crafted random baseline.

- l.281 An important point is to give precise numbers on how many of those protein-RNA complexes are part of ribosomal complexes. Out of the final 2022 pairs, how many of these are related to ribosomes? Some key figures on statistics (e.g. Fig. 2D) could be provided for both types separately in the sup. mat.

- Fig. 1: It is not clear how we go from 4173 complexes to 115085 interfaces. Does this mean there are ~30 interfaces per PDB file, on average? This could be described.

- Fig. 1: after filtering, only 3383 interfaces are retained. Why are so many interfaces filtered out? Is it due to RNA in contact being too small? In that case, should have it been considered as protein-RNA complex in the first place? This should also be described, even if with just one sentence.

- Fig 2B should also highlight the final set of 2022 pairs (with a different color for instance). This set is more important to highlight than the 6079 pairs, which are the only ones shown currently. I would also suggest plotting a contour density for the 3 sets rather than using points are those obscur the density completely.

Minor comments / typos

- l.154 the three tools MM-align, RNA-align and TM-align should have references.

- l.158 "out of over 444000 comparisons" -> one has to check all the material and methods to understand why it is not (977*977)/2-977/2 = 476776 comparisons. A short sentence with a reference to the "Structural alignment" part in the material and methods would help here.

- l.159 MMalign succeeds in aligning 207000 pairs -> why did it not succeed for the others? What are the criteria for a successful alignment? Is it that MMalign could not align at all, or that the TM-score is < 0.5?

- l.179 and fig.2: if I understand correctly: out of the 977 interfaces, 765 are kept and form 2022 pairs of interologs. This could be added to fig.2A to make things more clear for the reader.

- Fig S1E, S2, S3, S4 have no label for ordinate, the font and font size are not consistent intra-figures.

- figS4 is poorly presented.

- The authors also made available a website. I browsed it, and it is easy to use. One additional option could be, if possible, to enable the user to search for a specific PDB identifier from the home page. If one wants to see if a specific PDB structure has interologs I would rather type in the PDB code in a search bar to check if it belongs to any group. From what I understand it is currently only possible to search inside of a group.

- l. 538 I understand why clustering at 100% sequence identity for proteins, but why at 99% for RNAs?

- l. 567: why did FREESASA fail? Is it technical? 141 interfaces is a lot.

Reviewer #3: The main objective of the study summarized in the manuscript is to explore and characterize the evolution of protein-RNA interface. To this aim the authors generated a data base of interface-interface pairs (defined as interlogs) which provide a very good resource for the community.

Overall, based on a comprehensive analysis of the interlogs (divided to groups according to the degree of conservation, the authors conclude that in general protein-RNA interfaces are more conserved than protein-protein interfaces and that the most conserved contacts are the apolar contacts while hydrogen bond are the least conserved. While this conclusion is consistent with many previous studies it provides additional insights, which are supported by a relative large dataset. Finally, the authors built a user friendly and informative web server, which can be of great contribution to the protein-RNA community.

Below are my detailed remarks:

- As stated, the main aim of this work is to study protein-RNA interface evolution. Given that the majority of the interfaces are from the ribosome the authors correctly divide the results to ribosomal and non-ribosomal interfaces. However, given the importance of this grouping for the interpretation of the results, it is critical that the results of each group are discussed separately and preferably presented in a main figure. Moreover, I do not find that the violin plot showing the results summarized for all sub groups (based on conservation level) are of any value. What can one learn from looking at % of conserved contact when the level of conservation range between 10 to 90% ?

- The conclusions of this study are based on the analysis of the two major interface groups (ribosomal and non-ribosomal). Given that these results are suggested/expected to provide key features for modeling it is critical to show how/whether the relative conservation differs between interfaces coming from different RNA binding protein families (based on Rfam). As for example one would expect that the interfaces from RRM will differ from those that bind double strand RNA. Previous studies, as for example PMID: 25932908, have shown that the interfaces of the RNA bound to the different protein families have unique properties. It is thus critical that the author repeat their analyses upon grouping the interfaces based on the main RBP domain involved in the interface. As is, the authors should be much more cautious in the interpretations of the results and tone down their conclusions regarding the contribution of their study and its potential to be implemented for molecular modeling and docking. I suspect that the current analysis is not sufficient to provide useful scoring functions for protein-RNA modeling.

- The comparison between “non-conserved contacts” and “switching out of the interface” seems confusing to me, as to my understanding there is a strong dependency between the two and both are strongly dependent on the interface and interlog definitions, which were delineated by the authors.

- I find the following sentences either misleading or extremely over estimated and should be rephrased/toned down

1. “we focused on the analysis of protein-RNA interface evolution, aiming to unravel

relevant keys for structural modeling and prediction of protein-RNA complexes and for interpretation of multiple sequence alignments”

This sentence does not adhere to the results presented in this study.

2.“Numerous studies show that hydrogen bonds play a crucial role in protein-RNA interactions by contributing to the specificity and stability of the complexes and the function of RNA-protein complexes [45, 46]. “

The references cited are not suitable. There are many other key references that are clearly more relevant (several that are cited in the manuscript ) and should replace the ones cited.

3. “…as the ability to form H-bonds through the amino acid backbone does not vary with evolutionary mutations.”

This sentence is inaccurate and contradicts some of the message in the paper regarding the evolution of interfaces and changes in geometry that can surely affect the h-bonding via the protein backbone atoms.

4 “…and we evidenced increased contact conservation for evolutionarily more conserved interface regions. This latter feature is especially interesting since it does not require prior experimental knowledge of a protein-RNA interface structure.”

The observation that increased contact conservation for evolutionarily more conserved interface regions seems very trivial. Do the authors mean that there is increased contact conservation for evolutionarily more conserved surface regions? this is a very important message and critical for predicting protein-RNA interfaces.

5. The author claim that “we offer on protein-RNA interface evolution for a diverse range of protein and RNA families open new possibilities for tailoring scoring metrics to further improve protein-RNA specificity predictions” however, the paper they cite actually show that a general scoring function is not sufficient and family specific functions are required. This point is critical and should be emphasized.

**Have the authors made all data and (if applicable) computational code underlying the findings in their manuscript fully available?**

Reviewer #1: None

Reviewer #2: Yes

Reviewer #3: Yes

PLOS authors have the option to publish the peer review history of their article (what does this mean?). If published, this will include your full peer review and any attached files.

Reviewer #1: No

Reviewer #2: No

Reviewer #3: No
---

## [Decision Letter · Decision Letter 1]

18 Nov 2024

Dear Dr Andreani,

We are pleased to inform you that your manuscript 'Structural comparison of homologous protein-RNA interfaces reveals widespread overall conservation contrasted with versatility in polar contacts' has been provisionally accepted for publication in PLOS Computational Biology.

Best regards,

Rachel Kolodny

Academic Editor

PLOS Computational Biology

Arne Elofsson

Section Editor

PLOS Computational Biology

Feilim Mac Gabhann

Editor-in-Chief

PLOS Computational Biology

Jason Papin

Editor-in-Chief

PLOS Computational Biology

Reviewer's Responses to Questions

**Comments to the Authors:**

Reviewer #1: The revised manuscript is suitable for publication.

Reviewer #3: The authors have responded to all questions and remarks that were raised by this reviewer and have conducted the required revisions to the manuscript. I find that the manuscript has improved substantially and I have no further comments.

**Have the authors made all data and (if applicable) computational code underlying the findings in their manuscript fully available?**

Reviewer #1: None

Reviewer #3: Yes

PLOS authors have the option to publish the peer review history of their article (what does this mean?). If published, this will include your full peer review and any attached files.

Reviewer #1: No

Reviewer #3: No

---

## [Editor Report · Acceptance letter]

26 Nov 2024

PCOMPBIOL-D-24-00879R1 

Structural comparison of homologous protein-RNA interfaces reveals widespread overall conservation contrasted with versatility in polar contacts

Dear Dr Andreani,

I am pleased to inform you that your manuscript has been formally accepted for publication in PLOS Computational Biology. Your manuscript is now with our production department and you will be notified of the publication date in due course.

With kind regards,

Zsofia Freund
